# Mycobacterial Spindle Cell Pseudotumor Presenting as a Pancreatic Head Mass: A Case Report

**DOI:** 10.3390/pathogens14090889

**Published:** 2025-09-05

**Authors:** Frank A Cusimano, Tara Herrera, Douglas Brust, Elizabeth Montgomery, Sunil Amin, Folusakin Ayoade

**Affiliations:** 1Department of Medicine, Jackson Memorial Health System, University of Miami, Miami, FL 33136, USA; tara.herrera@jhsmiami.org; 2Department of Infectious Disease, University of Miami, Miami, FL 33136, USA; fxa375@miami.edu; 3Department of Infectious Disease, CAN Community Health, Cape Coral, FL 33602, USA; 4Department of Pathology, University of Miami, Miami, FL 33136, USA; 5Division of Gastroenterology and Hepatology, University of Miami, Miami, FL 33136, USA

**Keywords:** pseudotumor, mycobacterium spindle cell pseudotumor, spindle cell, histiocytes, mycobacterium, *Mycobacterium avium* complex, HIV, AIDS

## Abstract

Mycobacterial spindle cell pseudotumors (MSCPs) are rare lesions characterized by the proliferation of spindle-shaped histiocytes caused by mycobacterial infections. MSCPs have been reported in the lung, lymphatic system, and skin of immunodeficient patients. We present the case of a spindle cell pseudotumor of the pancreas in a 30-year-old male with advanced human immunodeficiency virus (HIV) infection, which led to biliary stricture, splenomegaly, chronic pancreatitis, portal hypertension, compression of the hepatic artery and portal vein, and ascites. This was the patient’s third mycobacterial infection diagnosis. The MSCP was diagnosed via endoscopic biopsy after two prior non-diagnostic biopsies of the pancreatic lesion. Following 18 months of tailored antimycobacterial therapy, the pancreatic mass resolved radiographically with normalization of liver tests and sustained clinical improvement, and there has been no relapse more than 8 months after treatment completion. This case highlights the presentation of an MSCP in a unique anatomic location not previously documented and the challenges encountered with diagnosis and management.

## 1. Introduction

Mycobacterial spindle cell pseudotumors (MSCPs) are rare benign, tumor-like lesions containing acid-fast mycobacteria. They are characterized by a local proliferation of spindle-shaped histiocytes on histopathology and are often misdiagnosed as soft tissue sarcomas or Kaposi’s sarcoma [1,2]. MSCPs typically occur in immunosuppressed patients, such as those living with HIV infection or those undergoing immunosuppressive therapies, as seen in transplant recipients, stem cell patients, individuals receiving radiation therapy, or patients on chemotherapy [3,4,5,6]. Infected patients are most commonly affected by *Mycobacterium avium* complex, *Mycobacterium tuberculosis*, *Mycobacterium chelonae*, or other mycobacterial species, although in ten previous cases, *No* organism was identified [1,3,7]. MSCPs are most frequently found in lymph nodes, skin, lungs, or brain [2,3,4,8,9,10,11,12,13]. Intra-abdominally, MSCPs have been reported in the spleen, liver, colon, and appendix [14,15,16,17,18]. To our knowledge, this is the first reported case of MSCP in the pancreas, highlighting the rare presentation of a disseminated mycobacterial infection in a patient with advanced HIV. Here, we discuss both the diagnostic challenges encountered in this case, as well as the challenges of managing multiple simultaneous non-tuberculous mycobacterial infections in a severely immunocompromised patient. Beyond its novelty, this case adds to clinical practice by (i) expanding the differential diagnosis of pancreatic masses in immunocompromised patients to include infectious etiologies, such as MSCP; and (ii) underscoring the importance of routinely sending biopsy material for mycobacterial culture and, when available, molecular testing to reduce diagnostic delay.

## 2. Case Report

### 2.1. Patient Background

A 30-year-old male presented with right upper quadrant abdominal pain that had worsened over several weeks. He has an extensive past medical history, including advanced HIV, diagnosed two years prior, with a recent cluster of differentiation 4 (CD4) count of 96 cells/μL and HIV viral load of <20 copies/mL. He has a history of bipolar disorder, ascites, hepatosplenomegaly, chronic pancreatitis, peptic ulcer disease, and *Mycobacterium avium complex* (MAC) immune reconstitution inflammatory syndrome (IRIS). This was associated with widespread lymphadenopathy, granulomatous hepatitis, and hypercalcemia with subsequent obstructing nephrolithiasis. His clinical course has been complicated. Since his HIV diagnosis, his clinical course has been marked by multiple infections including *Pneumocystis jirovecii* pneumonia, syphilis, oral and esophageal candidiasis, and disseminated MAC, which was identified on both blood cultures and tissue culture from an excisional supraclavicular (neck) lymph node biopsy. He subsequently had *Mycobacterium grossiae* followed by a third mycobacterium infection with *Mycobacterium abscessus*, isolated on a follow-up liver biopsy, all of which he was actively being treated for. His home medications include abacavir-dolutegravir-lamivudine, spironolactone, furosemide, lactulose, ondansetron, aripiprazole, fluoxetine, bupropion, trazodone, fluticasone, atovaquone, ethambutol, clarithromycin, and doxycycline.

### 2.2. Four Months Before Admission

Four months prior to admission, he presented with an acute elevation of his liver transaminases and associated intrahepatic and extrahepatic biliary duct dilatation. At that time, he was found on contrast-enhanced computed tomography (CT) of the abdomen and pelvis to have a cystic pancreatic head mass measuring 3.8 cm × 2.0 cm × 4.5 cm with mass effect on the common bile duct and main portal vein but without evidence of pancreatic necrosis. Follow-up CT of the abdomen and pelvis two weeks later showed the development of necrotic pancreatic cyst components with persistent mass effect, and severe narrowing to near occlusion of the main portal vein, with a retroperitoneal soft tissue component encasing the abdominal aorta, as well as splenic lesions. He had an endoscopic retrograde cholangiopancreatography (ERCP) performed, which showed a distal biliary stricture for which he had a sphincterotomy and a plastic stent placed. Endoscopic ultrasound (EUS) with biopsy of the pancreatic mass was performed twice with nondiagnostic histopathology showing histiocytes, pancreatic parenchyma with fibrosis, foreign body giant cells, and patchy acute inflammation. Findings were concerning for possible chronic pancreatitis without evidence of malignancy. On both occasions, tissue was not submitted for microbiologic (bacterial, mycobacterial, or fungal) culture or molecular testing.

### 2.3. Index Hospitalization

On admission, he presented with right upper quadrant abdominal pain and worsening ascites not controlled with diuretics. Laboratories were consistent with a biliary obstructive process (alkaline phosphatase 906 IU/L (normal range 40–130, international units (IU)/Liter (L)), aspartate transaminase (AST) 98 IU/L (normal range 10–40), alanine transaminase (ALT) 100 IU/L (normal range 0–41), total bilirubin of 4.2 mg/dL (normal range 0–1.2, milligrams/deciliter (mg/dL)). Initial paracentesis yielded 2.2 L of bland fluid, suggesting ascites from portal hypertension, serum ascites albumin gradient (SAAG) 2.4. Repeat contrasted abdomen and pelvis CT imaging now showed the necrotic pancreatic mass had increased to 6.8 cm × 4.5 cm with severe stenosis of the main portal vein, inferior vena cava, and hepatic artery (Figure 1A). The indwelling biliary stent showed a distal tip approximately 1 cm from the duodenal lumen. There was worsened intrahepatic and extrahepatic biliary ductal dilation, with the common hepatic duct measuring 1.9 cm compared to 1.2 cm on previous imaging and the right duct measuring 1.6 cm compared to 0.6 cm on prior imaging (Figure 1B,C). Results were concerning for migration of the common bile duct stent with worsening bile duct dilation and ascites.

The patient subsequently underwent an upper esophagogastroduodenoscopy (EGD) with EUS/ERCP for possible stent exchange and a repeat biopsy of the pancreatic head mass. Endoscopic ultrasound showed the ill-defined inflammatory appearing pancreatic head mass measuring 4.51 cm × 4.0 cm (Figure 2). A repeat fine-needle biopsy of the pancreatic mass was performed, and the biliary stent was exchanged (Figure 3). Again, tissue was not submitted for microbiologic culture or molecular testing. Two 10 Fr × 9 cm plastic stents with a single internal and external flap were placed in the right posterior and right anterior ducts. Twenty-four hours after stent exchange, the AST and ALT levels improved to 45 IU/L and 60 IU/L, respectively, alkaline phosphatase improved to 800 IU/L, and total bilirubin improved to 1.6 mg/dL. Despite improvement from the biliary obstruction, he suffered from recalcitrant ascites requiring daily paracentesis, which culminated in a 6-month hospital stay. A liver biopsy was performed on the same hospital stay due to the refractory ascites, which showed concentric fibrosis and irregular periportal fibrosis with possible early delicate bridging though no definitive cirrhosis. He was not deemed a candidate for surgical removal of the pancreatic head mass, so he underwent portal vein venoplasty and stent placement with interventional radiology. During admission, he also suffered from cholecystitis requiring a laparoscopic cholecystectomy, acute kidney injury, and an upper gastrointestinal bleed secondary to esophageal varices requiring banding ligation.

### 2.4. Pathology

The final surgical pathology report of the fine-needle biopsy of the pancreatic head mass resulted as a mycobacterial spindle pseudotumor. Histopathology was consistent with spindled histiocytes containing numerous acid-fast bacilli in both Periodic Acid-Schiff stain and acid-fast stain (Figure 4). No malignant neoplasm was seen.

### 2.5. Antimycobacterial Therapy and Outcomes

Infectious disease was consulted, and the patient was started on Amikacin 15 mg/kg intravenously (IV) daily, ethambutol 1200 mg (milligrams (mg)) oral daily, clarithromycin 500 mg oral twice daily (no inducible erm gene), and cefoxitin 2 g (grams (g)) IV every four hours. After a few weeks on this regimen, he developed a generalized maculopapular rash felt to be secondary to cefoxitin, which was subsequently changed to imipenem 1 g IV twice daily. He developed ototoxicity from Amikacin after approximately 9 months of therapy, which was also stopped. After a few lapses due to Omadacycline authorization delay, he was eventually transitioned to Omadacycline 300 mg oral daily, imipenem 1 g twice daily, and azithromycin 500 mg oral daily for an additional 9 months, which has since been completed. Table 1 provides a summary of the various treatment phases throughout his clinical course. Repeat liver biopsy after 1 year of anti-mycobacterial therapy showed improvement of the granulomatous disease. The patient was treated with a combination of intravenous agents and oral antibiotics for a total of 18 months and continues to follow with infectious disease as an outpatient. His pancreatic head lesion completely resolved on follow-up CT imaging with normalization of his liver enzymes and significant improvement of his ascites. He has since discontinued treatment for his disseminated mycobacterial infection for over 8 months, has remained compliant with his antiretroviral therapy, and maintained an undetectable HIV RNA viral load and a CD4 count above 200 cells/mm^3^ (cubic millimeters (mm^3^)). He has not required a paracentesis for ascites for over a year and continues to do well off anti-non-tuberculous mycobacterial (anti-NTM) therapy. He has been able to gain weight and, to date, there has been no relapse of any of his mycobacterial infections.

## 3. Discussion

Here, we present a case of portal hypertension and recalcitrant ascites due to a disseminated NTM infection involving the liver, and a mycobacterial spindle cell pseudotumor presenting as a pancreatic head mass. To the best of our knowledge, this is the first reported MSCP in the pancreas.

MSCP may be mistaken for soft tissue sarcomas or Kaposi’s sarcoma due to considerable histologic overlap [2]. Diagnosis requires histopathological evaluation demonstrating classical spindle-shaped cells on microscopy and the use of special stains of the biopsied tissue of interest, typically Ziehl–Neelsen staining for acid-fast mycobacteria [19]. In our case, the monotonous proliferation of spindle-shaped histiocytes arranged in a storiform pattern, absence of significant cytologic atypia or mitotic activity, and the demonstration of abundant AFB-positive bacilli within histiocytes favored MSCP over sarcoma. The clinical context of advanced immunosuppression and concurrent NTM infection further supported this diagnosis. MSCP/NTM therapy should be initiated when there is clinical, radiographic, histopathologic, and/or microbiologic evidence of disease and should be individualized to species and drug susceptibility results [20].

While diagnosis requires histopathological evaluation, treatment is best guided by cultures, which allow for antimicrobial susceptibility testing to guide pathogen-targeted therapy. Despite two prior biopsies, tissue from the pancreatic mass was not submitted for mycobacterial culture or molecular testing, which likely delayed definitive etiologic identification and limited the clinicians’ opportunity to use more narrowed therapeutic options. When microbiologic diagnosis is feasible, biopsy specimens should be allocated for AFB smear and culture, fungal and bacterial cultures, and, where available, molecular testing (e.g., mycobacterial polymerase chain reaction (PCR) and broad-range 16s ribosomal ribonucleic acid (RNA) sequencing) that can be performed on tissue. In our patient, identification of *Mycobacterium grossiae* (identified via 16s RNA sequencing) on liver tissue and subsequent recovery of *Mycobacterium abscessus subspecies abscessus* on a later liver biopsy guided selection of a regimen inclusive of both NTM species.

Table 2 provides a review of documented MSCPs across organ sites, host status and outcome. Our case is only one of two with more than one documented NTM organism identified [4]. Most occurred in immunocompromised hosts (acquired immunodeficiency syndrome (AIDS), autoimmune disorders on immunomodulatory therapies, post-transplant) and were due to nontuberculous mycobacteria—most often MAC—with less frequent cases due to *Mycobacterium tuberculosis* or fastidious NTM (e.g., *Mycobacterium haemophilum*, *Mycobacterium simiae*). Rare immunocompetent presentations have been reported in the CNS [10]. There were several reports of intraabdominal MSCPs, but none in the pancreas, as with our patient [14,18]. A portion of MSCPs of other organ sites were diagnosed near to or after the death of the patient [3,11,14,18].

MSCPs can be managed medically, surgically, or with a combination of medical and surgical approaches. One review documented more significant success with antimicrobial therapy compared to no treatment or surgical resection alone, even though the group mortality was as high as 25% [1]. Success in this review was defined as a clinical and radiological response while on antimycobacterial therapy [1]. Medical management typically involves at least a 3-drug regimen, often with a combination of oral and parenteral agents, all tailored to the cultured mycobacterial organism and susceptibilities [20]. Treatment durations are also quite long. A review of MSCPs of the CNS documented a range of 6 months to up to 2 years of treatment [21]. In our review of documented MSCPs, for those patients who successfully started antimycobacterial therapy, there was eventual documented improvement or resolution of both symptoms and lesions, and these cases often required at least partial surgical resection [8,13,21]. In our case, the patient was not deemed a candidate for surgical resection of the MSCP. The mass effect it produced (biliary obstruction and severe portal hypertension) was managed with venoplasty/stenting, and complete radiographic and clinical response was achieved with prolonged antimycobacterial therapy.

In addition to the time burden involved with treatment and the number of agents required for treatment, anti-mycobacterial regimens are not benign. Chronic exposures to such agents come with the risk of potential side effects as experienced with our patient. Our patient developed ototoxicity due to amikacin, as well as a drug-mediated rash from exposure to cefoxitin, leading to several changes in his regimen for disseminated mycobacterial infection throughout his treatment course.

This case highlights the unique nature of immunodeficient patients as he was previously infected with not one, but three mycobacterial infections. In the appropriate clinical setting, the possibility of multiple simultaneous NTM infections should be considered, especially when treatment directed against the identified NTM appears to be ineffective [22]. Although our patient had close follow-up after his HIV diagnosis and was promptly placed on appropriate HIV treatment, he still developed significant complications, diagnostic and management challenges, and some omissions with his care. When a microbiologic diagnosis is possible or in consideration, providers should endeavor to include microbiological analysis (Gram-stain, bacterial, mycobacterial, fungal cultures, or molecular testing) as part of the diagnostic workup of biopsied tissue samples. For patients with HIV/AIDS (acquired immunodeficiency syndrome) or who are on immunosuppressive therapies, a broad differential diagnosis must be considered for pancreatic masses or cystic lesions. For such and similar lesions, biopsy remains the gold standard for diagnosis, and repeat biopsy may be needed at times for definitive diagnosis. Treatment of MSCP, especially when they involve body viscera, could be challenging as they often produce mass “tumor-like” effects on surrounding structures, as in our case [1,4,14]. Recalcitrant ascites in our patient was thought to be secondary to portal vein stenosis, which improved with portal vein venoplasty/stenting and shrinking of the pancreatic mass by anti-mycobacterial therapy.

## 4. Conclusions

Observation of spindle cells on tissue histopathology of mass lesions—especially in immunocompromised patients—should prompt consideration of mycobacterial infection and additional AFB staining. Routine submission of biopsy material for mycobacterial culture and consideration of molecular assays can shorten time to etiologic diagnosis and optimize therapy. Future research should focus on the epidemiology of MSCP, diagnostic strategies (including when to deploy molecular testing), treatment efficacy, and outcomes among people with HIV/AIDS.

## Figures and Tables

**Figure 1 pathogens-14-00889-f001:**
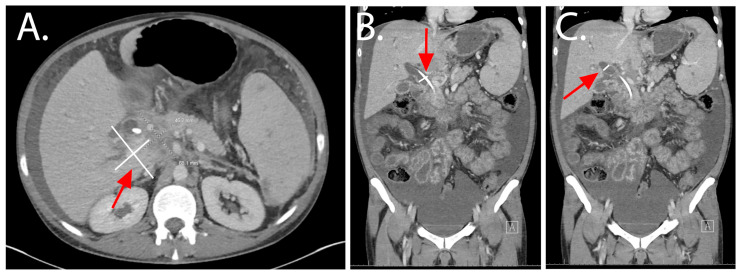
Computed Tomography of abdomen and pelvis: (**A**) Ill-defined infiltrative hypodense mass of the pancreatic head with peripancreatic inflammation extending to the porta hepatis measuring approximately 6.8 cm × 4.5 cm with severe stenosis of the main portal vein, inferior vena cava and hepatic artery. (**B**,**C**) Intrahepatic and extrahepatic biliary ductal dilation with the common hepatic duct measuring 1.9 cm compared to 1.2 cm on previous imaging and the right duct measuring 1.6 cm compared to 0.6 cm on prior imaging. White lines represent the measurements, and red arrows are provided to aid in location.

**Figure 2 pathogens-14-00889-f002:**
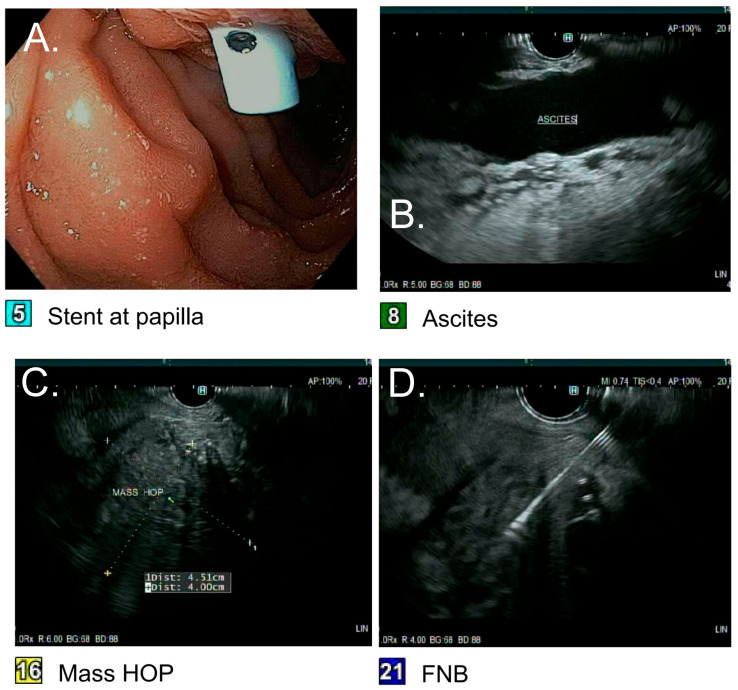
Endoscopic ultrasound with fine-needle biopsy of the pancreatic mass: (**A**) Visualization of the stent at the papilla placed weeks prior. (**B**) Ultrasound imaging of abdominal ascites. (**C**) Endoscopic ultrasound view of the pancreatic head mass (Head of Pancreas (HOP)) measuring 4.51 cm × 4.00 cm. (**D**) Fine-needle biopsy (FNB) of the pancreatic head mass.

**Figure 3 pathogens-14-00889-f003:**
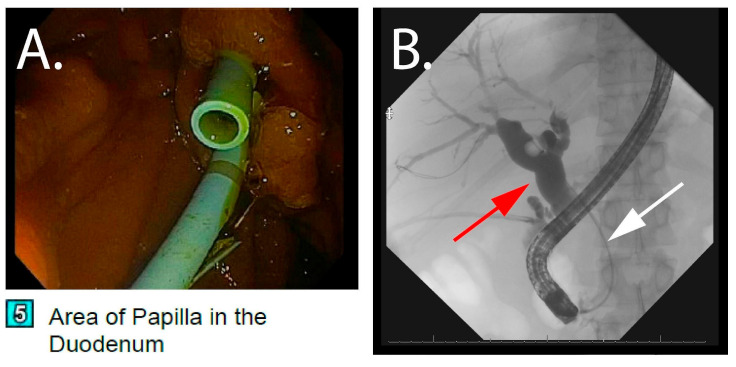
Endoscopic retrograde cholangiopancreatography with stent exchange: (**A**) Visualization of the plastic stents at the papilla during stent exchange. (**B**) Fluoroscopic imaging showing biliary dilatation (red arrow) and distal stricture (white arrow) from compression from the pancreatic head mass.

**Figure 4 pathogens-14-00889-f004:**
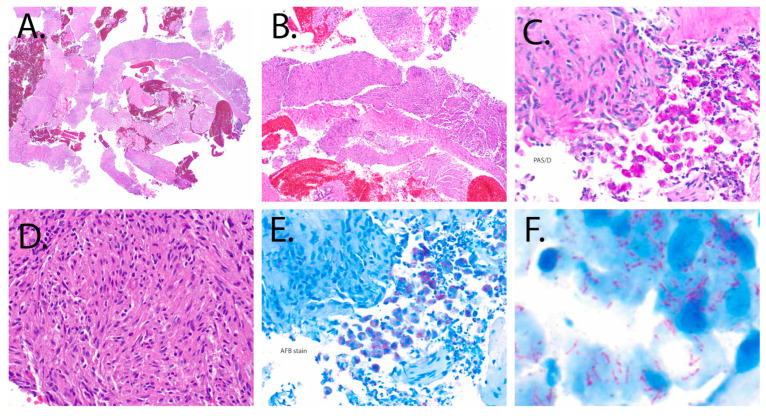
Pathological findings from endoscopic fine-needle biopsy: (**A**) Low-power photomicrograph (20×) hematoxylin and eosin (H&E) stain showing cellular cores of “neoplastic”-appearing cells admixed with fibrous tissue, duodenal epithelium, and blood. (**B**) Medium-power magnification (100×) H&E stain of the same tissue section seen previously illustrates a storiform pattern, characterized by a monotonous proliferation of spindle-shaped cells. (**C**) Periodic Acid-Schiff (PAS) stain (digested, 200×) highlights spindled histiocytes containing numerous intracytoplasmic bacilli (dark pink). (**D**) High-power magnification (200×) demonstrates a spindle cell histiocytic population intermixed with sparse lymphocytes. The spindled histiocytes exhibit eosinophilic cytoplasm, elongated epithelioid nuclei with smooth contours, vesicular chromatin, and inconspicuous nucleoli. (**E**,**F**) High-power magnifications (200× and 600×) of an acid-fast bacilli (AFB) immunohistochemical stain reveals abundant acid-fast bacilli (pinkish-purple), supporting the diagnosis of a mycobacterial spindle cell tumor.

**Table 1 pathogens-14-00889-t001:** Summary of antimycobacterial treatment and adverse events after MSCP diagnosis.

Treatment Phase	Regimen	Approximate Duration	Reason for Change/Key Events
Initial	Amikacin, Ethambutol, Clarithromycin, Cefoxitin	First few weeks	Maculopapular rash attributed to cefoxitin
Phase 2	Amikacin, Ethambutol, Clarithromycin, Imipenem	Subsequent 9 months	Ototoxicity from amikacin; amikacin discontinued
Phase 3	Omadacycline, Imipenem, Azithromycin	Additional 9 months	Completed to total 18 months’ therapy; radiographic and clinical resolution

**Table 2 pathogens-14-00889-t002:** Review of published mycobacterial spindle cell pseudotumor (MSCP) cases across organ sites.

Reference (Year)	Organism Identified	Host (Status)	Site	Outcome
Current report	*Mycobacterium avium complex* (blood + supraclavicular lymph node culture)*, Mycobacterium grossiae* (liver biopsy)*, Mycobacterium abscessus* (liver biopsy)	Advanced HIV/AIDS; CD4 nadir < 50	Pancreas	Mass effect resolved on prolonged multi-drug therapy; biliary/portal HTN complications improved
Suster et al., 1994 [14]	Nontuberculous mycobacteria (species not reported)	Advanced HIV/AIDS	Localized to spleen at autopsy	Died (Other cause-CNS toxoplasmosis; MSCP incidental at autopsy)
Sekosan et al., 1994 [3]	*Mycobacterium tuberculosis*	Solid-organ transplant recipient	Pulmonary	Died within 4 days of starting Tuberculosis treatment
Chesdachai et al., 2020 [4]	*Mycobacterium avium–intracellulare complex*	Liver transplant recipient	Pulmonary (masses), later also found on colonoscopy	Not reported
Morrison et al., 1999 [13]	*Mycobacterium avium–intracellulare complex*	HIV-negative; chronic steroids for sarcoidosis	Left frontal lobe, both parietal regions, and the right cerebellar hemisphere	Lesions resolved with surgical resection + anti-NTM therapy: Amikacin, clarithromycin, ethambutol + rifampin→ azithromycin, ethambutol + rifampine; amikacin was discontinued because of worsening ataxia. No comment on the duration of therapy.
Phowthongkum et al., 2008 [8]	*Mycobacterium haemophilum + Mycobacterium simiae (mixed)*	Advanced HIV/AIDS	Multiple lesions in the left cerebral peduncle and medulla	Partial surgical excision of the left medulla + anti-NTM therapy: Isoniazid, rifampin, pyrazinamide, ethambutol, + clarithromycin. No comment on the duration of therapy. Documented improvement with some residual neurological deficits.
Poyuran et al., 2021 [10]	Not reported	Immunocompetent adult	Intracranial lesion	Not reported
Basílio-de-Oliveira et al., 2001 [18]	NTM (species not determined)	Advanced HIV/AIDS (autopsy case)	liver, lymph nodes, spleen, skin, large and small intestine; also with disseminated histoplasmosis	Died (other cause-overwhelming sepsis)
Rahmani et al., 2013 [11]	*Mycobacterium avium (PCR)*	Liver transplant recipient	Cutaneous	Died (other cause- multiorgan failure)
Shiomi et al., 2007 [12]	*Mycobacterium intracellulare*	SLE (on prednisolone and azathioprine), insulin-dependent diabetes mellitus	Cutaneous	Not reported
Duong et al., 2024 [21]	*Mycobacterium avium (blood + cerebrospinal fluid)*	Advanced HIV/AIDS	Intradural spinal cord mass at L3–4	Azithromycin, ethambutol, rifampin → azithromycin, ethambutol, rifabutin for >1 year with ART

Abbreviations: NTM, nontuberculous mycobacteria; HIV, human immunodeficiency virus; AIDS, acquired immunodeficiency syndrome; SLE, systemic lupus erythematosus; ART, antiretroviral therapy; HTN, hypertension; MSCP, mycobacterial spindle cell tumor; CNS, central nervous system.

## Data Availability

No new data were created or analyzed in this study. Data sharing is not applicable to this article.

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
