# Peer review of "Mycobacterial Spindle Cell Pseudotumor Presenting as a Pancreatic Head Mass: A Case Report"

_pathogens, 2025, doi:10.3390/pathogens14090889_

Round 1
Reviewer 1 Report
Comments and Suggestions for Authors
Dear Authors,
The manuscript titled "Mycobacterial Spindle Cell Pseudotumor Presenting as a Pancreatic Head Mass: A Case Report" is an interesting case report. Please find my comments below:
Line 52: Have you identified the bacterium (Mycobacterium avium complex) in blood samples?
Line 55: The phrase could be more precise: His clinical course has been complicated
Line 151: Could you clarify whether controlling Mycobacterial Spindle Cell Pseudotumor (MSCP) has the potential to enhance the effectiveness of antiretroviral therapy in HIV patients? Given that MSCP can occur at different stages of HIV disease, and infections might remain silent for some time, would you recommend including treatment for MSCP in the regimen for all HIV patients? If so, please consider reflecting this clarification in the manuscript.
Figure 4: Further clarification of the cellular compartments in the figure, as well as an explanation of the stains and staining procedure, is needed.
Author Response
- We have clarified that the initial diagnosis of disseminated MAC was made through both blood cultures and tissue culture from an excisional supraclavicular lymph node biopsy. Where: Case Report → Patient background (highlighted- line 69).
- For more precise wording, we added a sentence stating that the patient’s “clinical course has been complicated”. In addition we rephrased the following sentence to state that his “clinical course has been marked by multiple infections …” followed by a description of those infections.
Where: Case Report → Patient background (highlighted- line 66). - In regards to the question of ART effectiveness & universal MSCP therapy: There is not enough evidence, to the authors’ knowledge, that universal NTM treatment should be implemented for patients living with HIV, in an attempt to enhance the effectiveness of ART. I added a sentence stating, “MSCP/NTM therapy should be initiated when there is clinical, radiographic, histopathologic and/or microbiologic evidence of disease and should be individualized to species and drug susceptibility result.”. We hope this emphasizes the need for objective evidence of infection as the prompt for treatment, irrespective of the need for immune reconstitution.
Where: Discussion → line 217 (highlighted)
- Descriptions of specific stains were added to photos A and B in Figure 4, as well as a color description to help clarify what in the biopsy that readers are looking at.
Reviewer 2 Report
Comments and Suggestions for Authors
This is a well-documented and compelling case report detailing a rare presentation of mycobacterial spindle cell pseudotumor (MSCP) in the pancreas of an immunocompromised HIV-positive patient. The case is both clinically significant and novel, particularly as this appears to be the first documented case of pancreatic MSCP. The manuscript is generally well-organized, methodologically sound for a case report, and addresses an important clinical challenge—atypical presentations of disseminated mycobacterial infections in patients with advanced HIV/AIDS.
However, prior to publication, revisions are necessary to improve clarity, correct omissions, and enhance the scientific rigor of the discussion. These are outlined below.
Major Comments
The presentation of MSCP in the pancreatic head is genuinely novel and well justified with literature support, but please explicitly state in the Introduction how this case changes or adds to current clinical knowledge and why it deserves publication (e.g., impact on diagnostic pathways, management algorithms, or awareness of diagnostic delays).
Despite two prior biopsies, no microbiologic cultures were obtained from the pancreatic mass. While this limitation is acknowledged, its implications are not thoroughly discussed. Please provide a more critical reflection on how this affected diagnosis and treatment selection. Discuss whether PCR or molecular diagnostics could have improved pathogen identification retrospectively.
The treatment narrative is detailed but could be better organized.
Summarize the antimycobacterial regimens and adverse events (drug, duration, reason for change, toxicity) for clarity and to avoid overwhelming the reader with text.
The discussion references similar lesions in spleen, colon, brain, etc. However, the literature review is somewhat superficial.
Please include a brief comparison table of published MSCP cases (location, organism, host status, outcome), highlighting how the pancreatic case differs.
There is no mention of patient consent for publication.
Please add a formal statement confirming written informed consent was obtained.
Minor Comments
Terminology Consistency:
Use consistent terminology throughout (e.g., "non-tuberculous mycobacteria" vs. "nontuberculous").
Acronym Overuse:
Avoid overloading the text with repeated acronyms. Use full terms where appropriate.
Grammar/Typo Corrections:
“presnting” shoul de “presenting” in the title (corrected).
Rephrase passive constructions in places for clarity.
Conclusion and Recommendation
This is an important and well-prepared case report with significant educational and clinical value, particularly in infectious diseases, HIV medicine, and gastroenterology. However, to meet publication standards, the manuscript requires moderate revisions to address:
- Incomplete microbiologic confirmation,
- Organization of therapeutic data,
- Enhanced discussion and literature context,
- Explicit consent and ethics clarification.
Author Response
- We now explicitly state the novelty of this case and its clinical implications.
Where: Introduction → line 50 (highlighted). - In an attempt to more deeply reflect on the lack of pancreatic tissue cultures and the role of molecular diagnostics, the following revisions were made:
Revision:- In the Case Report section, we explicitly note that the two prior pancreatic biopsies were not submitted for culture/molecular testing- lines 96, 129 (highlighted).
- In the Discussion section, we added a statement that the lack of tissue samples for culture and molecular testing likely caused a diagnosis delay and limited the opportunity to use more narrowed therapies- starting at line 224 (highlighted).
- In the Conclusion section, we added a statement highlighting the importance of sending samples for culture and if need, to utilize molecular testing as well (highlighted)
- We have re‑structured Case Report into subsections: Patient background, Four months before admission, Index hospitalization, Pathology, and Antimycobacterial therapy and outcomes.
- We added a time range for the onset of adverse effects after starting anti-NTM therapy- Line 169. Additionally we added a table summarizing the treatment narrative, length of continuation, and reason for drug discontinuation (Table 1).
- We added a statement that “Written informed consent was obtained from the patient for publication of this case report and accompanying images.” (highlighted- line 311).
Where: Informed Consent Statement. - We added a comparison table (Table 2) of published MSCP cases to highlight how our case differs from current published case reports. We also used this to enhance our discussion of the literature (See highlighted portions in the Discussion section: 4th paragraph- starting at line 234, and 5th paragraph -starting at line 246).
Reviewer 3 Report
Comments and Suggestions for Authors
This case report presents what appears to be the first documented instance of a mycobacterial spindle cell pseudotumor (MSCP) occurring in the pancreas of an HIV-positive patient. The patient experienced complications including portal hypertension, biliary obstruction, and refractory ascites, which were ultimately resolved with prolonged combination antimycobacterial therapy.
Consider making clinical timeline in the case description clearer, for example, breaking this section down into smaller paragraphs
The introduction notes that MSCP are often misdiagnosed as soft tissue sarcomas or Kaposi’s sarcoma. Consider adding a discussion to explicitly mention how the diagnosis was distinguished from these mimickers
The abstract currently describes the case and its challenges but doesn’t talk about the outcome, consider adding one sentence about the outcome
“Elizabeth Montgomery MD 4,, Sunil Amin MD 5” contains a double comma
“cells/mm3 (cubic meters)” should read “cubic millimeters” (mm3) or simply be expressed as cells/μL
“NTM” (nontuberculous mycobacteria) is used in the discussion without earlier definition in the main text
Author Response
- We have added a sentence addressing how MSCP can be a Kaposi sarcoma mimic. We also discuss the histologic features that aid in the diagnosis of infection, instead of malignancy.
Where: Discussion → 2nd paragraph starting at line 208
- A sentence speaking to the patient outcome was added to the abstract - Line 22
- “cells/mm3 (cubic meters)” was changed to “cubic millimeters” (mm3)- Line 180
Reviewer 4 Report
Comments and Suggestions for Authors
This present study aims to describe the first reported case of a mycobacterial spindle cell pseudotumor (MSCP) occurring in the pancreas, and to underscore the diagnostic and therapeutic difficulties posed by MSCP, particularly in highly immunocompromised patients with concurrent infections caused by multiple non-tuberculous mycobacterial (NTM) species. This study offers novel insights in several key aspects: (i) It presents the first documented case of pancreatic MSCP, representing a unique addition to existing literature on NTM-associated pseudotumors; (ii) The detailed clinical account of an HIV-positive patient with triple mycobacterial coinfection enhances understanding of diagnostic and management complexities in such cases; (iii) It underscores significant treatment challenges in MSCP, including extended multidrug therapies and their associated toxicities. The key implications of this study: (i) By raising awareness of rare MSCP presentations, this work may help prevent clinical misdiagnosis (particularly confusion with malignancies or sarcomas); (ii) The case underscores the vital yet often neglected need for microbiological culture of biopsy specimens; and (iii) The findings contribute to the evolving diagnostic approach for pancreatic masses in immunocompromised patients, where such lesions are frequently mistaken for neoplastic processes.
Author Response
- Thank you for your feedback. We used it to strengthen our argument for the case’s novelty and contribution to the current clinical space.